# Tribological Properties of Ni/Cu/Ni Coating on the Ti-6Al-4V Alloy after Annealing at Various Temperatures

**DOI:** 10.3390/ma13040847

**Published:** 2020-02-13

**Authors:** Jinheng Luo, Nan Wang, Lixia Zhu, Gang Wu, Lifeng Li, Miao Yang, Long Zhang, Yongnan Chen

**Affiliations:** 1CNPC Tubular Goods Research Institute, Xi’an 710017, China; luojh@cnpc.com.cn (J.L.); zhulx@cnpc.com.cn (L.Z.); 15191899113@163.com (G.W.); lilifeng004@cnpc.com.cn (L.L.); 2CNPC Key Laboratory for PetroChina Tubular Goods Engineering, Xi’an 710077, China; 3School of Materials Science and Engineering, Chang’an University, Xi’an 710064, China; wangnanchd@163.com (N.W.); m18789475320@163.com (M.Y.); zhanglongsir@163.com (L.Z.)

**Keywords:** Ti-6Al-4V alloy, Ni/Cu/Ni coating, phases transitions, tribological properties, wear mechanism

## Abstract

Diffusion reaction was a crucial route to enhance the wear resistance of Ti-6Al-4V alloys surface. In this work, the Ni/Cu/Ni composite layers were fabricated on the surface of Ti-6Al-4V alloy by electroplate craft, and then different annealing temperatures were applied to further optimize its tribological properties. The diffusion behaviors at various temperatures were systematically analyzed to reveal the physical mechanism of the enhanced tribological properties of the coatings. It was demonstrated that Cu_x_Ti_y_ and Ni_x_Ti_y_ intermetallic compounds with high hardness and strength were produced in the Ni/Cu/Ni coating, which acted as the reinforcing phases and improved the microhardness, reduced the friction coefficient, and lessened the wear rate. Specially, this effect reached the maximum when the annealing temperature was 800 °C, showing excellent wear resistance. This work revealed the relationship between annealing temperatures and tribological properties of the Ni/Cu/Ni coating, and proposed wear mechanism, aiming to improve the surface performance of Ti-6Al-4V alloy by appropriate diffusion behavior.

## 1. Introduction

Ti-6Al-4V alloys have the advantages of high specific strength, strong corrosion resistance, and good biocompatibility, which have been widely used in aerospace, marine development, and biomedical fields [1,2,3]. However, in some engineering applications, titanium alloys usually exhibit poor tribological properties, such as high friction coefficient, severe adhesive wear, low plastic shear resistance, weak work hardening ability, and brittle oxide film on the surface [4,5], which greatly limit their application as friction components. Aydin et al. [6] suggested that the performance of matrix materials could be effectively reinforced by the thermal diffusion behavior between different metals. Yao et al. [7] prepared the copper layer on the Ti-6Al-4V alloy, exploiting diffusion between copper and titanium atoms, which can greatly improve the surface hardness and wear resistance. In addition, multilayer diffusion had been designed to form a variety of intermetallic compounds in coating by adding interlayer materials, optimizing the mechanical performance of the coating [8,9].

As well known, the atoms in the surface coating will occur a diffusion reaction and form the diffusion layer with a certain thickness during annealing [6,10,11]. Hu et al. [10] insisted that the Ni-Ti diffusion layer was mainly composed of the NiTi_2_, Ni_3_Ti, and NiTi phases and hence it showed high hardness. Aydın et al. [6] proved that the formed Cu_x_Ti_y_ intermetallic compounds during annealing process could apparently improve surface properties because of its high hardness. It can be concluded that the improvement of hardness in the diffusion layer was strictly related to the intermetallic compounds, which can cause the wear mechanism transform to micro-adhesive wear from the adhesive wear, thereby improving the surface tribological properties. More importantly, the diffusion behavior of Ni and Ti atoms could be remarkably adjusted by the annealing temperatures [11]. Xia et al. [12] accurately analyzed the relationship between annealing temperatures and the properties of Ti-6Al-4V alloy coatings, and the results confirmed the metallurgical bonding could be easily formed to strengthen its surface properties. Shen et al. [13] found that the depth of diffusion layer and the type of intermetallic compounds will be changed with the rise of annealing temperatures, which could observably improve the surface capability of the coatings. The above studies showed that the surface tribological properties of Ti-6Al-4V alloys could be improved by appropriate annealing of dissimilar metals.

However, the relationship between multi-layer diffusion behaviors, surface tribological properties, and annealing temperatures were still indistinct. More importantly, a variety of intermetallic compounds may be dispersed in the diffusion layer without damaging the plasticity of the coating, which was essential for the study of wear mechanisms. Relevant research [2] had been done on Cu/Ni coating before, and a diffusion layer with certain properties has been obtained on the surface of the Ti-6Al-4V alloy. However, it was still not satisfactory in terms of the tribological properties. In this work, the diffusion behaviors and tribological properties of the Ni/Cu/Ni coating after annealing at various temperatures 600 °C, 700 °C, and 800 °C were studied. The formation mechanism of intermetallic compounds and the wear mechanism of diffusion layer were analyzed in detail. The purpose of this work was to study the diffusion layer evolution and wear resistance of the Ni/Cu/Ni coatings, obtaining the improved coating with a certain anti-wear effect by adjusting the annealing temperatures.

## 2. Experimental Procedure

### 2.1. Materials Synthesis

The commercial Ti-6Al-4V alloy was adopted for this study, and the chemical composition (wt.%) was 88.2Ti, 6Al, 4V, 0.6Fe, 0.5Mn, 0.4Si, and 0.3Zn. The electroplated Ni/Cu/Ni composite coatings were synthesized from the traditional electroplate method, the plating sequence of the Ni/Cu/Ni composite coating was first nickel plating for 10 minutes, then copper plating for 20 minutes, finally nickel plating for 10 minutes, and the detail electrodeposition conditions and compositions of the bath were summarized in Table 1. The flow diagram of sample preparation process was shown in Figure 1a, and the macro picture of sample obtained at each step of the preparation process was shown in Figure 1b, showing the macro change process of sample. The schematic diagram of the cross section and surface of the prepared sample were shown in Figure 1c. It could be evidenced the structure of the Ni/Cu/Ni coating on the Ti-6Al-4V substrate and the surface was uniform and dense. Then, the annealing process was performed in a vacuum furnace (OTF-1200X, MTI-Kejing, Hefei, China), setting the vacuum level of 1 × 10^−2^ Pa. The plating samples were annealing at 600 °C, 700 °C, and 800 °C for 3 h, respectively, after that, cooled down to 25 °C in the furnace.

### 2.2. Characterization after the Thermal Diffusion

The microstructure analysis of the annealing samples was detected using a scanning electron microscopy (SEM, Hitachi-S4800, Hitachi Limited, Tokyo, Japan). The thermal diffusion behaviors of Cu, Ni, and Ti atoms were interpreted using an energy dispersive spectroscopy (EDS, Hitachi-S4800, Hitachi Limited, Tokyo, Japan). The phases structures of the coatings were identified by X-ray diffraction (XRD, XRD D/M2500, Bruker, Karlsruhe, Germany) with Cu Ka radiation (0.154 nm wavelength) at tube voltage of 40 kV, current of 40 mA, an 8.0°/min scanning speed, a 20–80° of 2θ range.

### 2.3. Tribological Properties

The surface roughness and microhardness of the coating surface after annealing were measured by a roughness tester (SJ-210, Mitutoyo, Kawasaki, Japan) and microhardness tester (HV-1000A, Huaying, Laizhou, China), respectively. For data validity, five tests were performed and then averaged. The wear resistance of the coatings was assessed with a ball-on-plate wear tester (MMQ-02G, Yihua, Jinan, China) using a 6 mm GCr15 ball counterpart. A constant load of 3 N was applied normally to the sample under non- lubricated condition at room temperature. The abrasion resistance tests were performed on with a circular track of 3 mm in diameter, a rotational speed of 100 r/min and a total sliding distance of 37.70 m. The worn morphologies of the coatings were explained using scanning electron microscopy (SEM, Hitachi-S4800, Hitachi Limited, Tokyo, Japan) and Laser Confocal Microscope (LCM, Olympus OLS5000, Olympus Corporation, Tokyo, Japan) to show the worn mechanism.

## 3. Results and Discussion

### 3.1. The Diffusion Behaviors and Phases Transitions 

As shown in Figure 2a, the morphology and composition of Ni/Cu/Ni layers on Ti-6Al-4V alloy after annealing were researched by cross-sectional observation with EDS examination. The element distribution of Cu, Ni, and Ti in diffusion layers was clearly displayed. Obviously, the Cu, Ni, and Ti atoms hardly occurred diffusion behavior without annealing, where the interfaces in the Ni/Cu/Ni layers were obvious with a clear outline. However, these interfaces gradually became blurred and disappeared with the increasing of annealing temperatures, which indicated that diffusion behaviors between Cu, Ni, and Ti atoms intensified step by step. It was noteworthy that the diffusion layer with a certain thickness was formed between Ni/Ti interfaces, which was distinctly different from Ni layer and Ti-6Al-4V substrate. Moreover, the higher the annealing temperature was, the thicker the diffusion layer was (Figure 2b). Actually, Ti atom had a higher diffusion rate than Ni atom during the annealing process, which resulted in a large number of Kirkendall voids on the Ti side [14]. Furthermore, as the annealing temperatures increased, the voids will connect with each other and then coalesce into large gaps, forming the Kirkendall diffusion channels [15,16]. Finally, they would become a fast diffusion channel for all atoms (Figure 2c,d).

Figure 3 displayed the elements line scan diagrams across the coatings from its substrate to the top after annealing. It can be seen that without annealing, the interface of each layer was clear, and no atomic diffusion occurred (Figure 3a). With the increase of annealing temperatures, the interdiffusion between Cu, Ni, and Ti atoms intensified, gradually forming a continuous one. In addition, the interdiffusion ability of the element in the Ni layer was better than that in the Cu layer. As shown in Figure 3b–d, there were fewer Ni and Ti atoms in the Cu layer. In the Ni layer, Cu and Ti atoms were fully diffused and the diffusion reaction occurs, forming a series of intermetallic compounds and solid solutions. 

Figure 4a showed the Ni-Cu solid solution and the intermetallic compounds of Ni_x_Ti_y_ and Cu_x_Ti_y_ in the coatings when the annealing temperatures increased from 600 to 800 °C. According to the Ni-Cu equilibrium diagram, the α(Cu, Ni) solid solution (Cu_3.8_Ni) was mainly formed in the annealed coatings. For the Ni-Ti binary system, the following reactions (1)–(3) occurred after the diffusion of Ni and Ti atoms, and the Ni_3_Ti, NiTi_2_, and NiTi phases were formed, respectively [17,18]:(1)$$3\mathrm{Ni}+\mathrm{Ti}={\mathrm{Ni}}_{3}\mathrm{Ti}$$
(2)$$\mathrm{Ni}+2\mathrm{Ti}={\mathrm{NiTi}}_{2}$$
(3)Ni+Ti=NiTi

In order to reveal the formation mechanism of Ni_x_Ti_y_ intermetallic compounds, the Gibbs free energies of the above reaction were summarized in Figure 4b. The Gibbs free energies of Ni_3_Ti and NiTi_2_ phases were substantially lower than that of NiTi phase, which illustrated that the reaction (1) and reaction (2) were more likely to occur than reaction (3). Bastin et al. [14] found that the nucleation and growth of NiTi phase caused a sharp decrease in NiTi_2_ phase and Zhou et al. [19] also confirmed that the growth of NiTi phase considerably consume Ni_3_Ti and NiTi_2_ phases. According to previous analysis, the nucleation and growth of NiTi phase mainly depend on the Ni_3_Ti and NiTi_2_ phases. That is, the nucleation of the Ni_3_Ti and NiTi_2_ phases was earlier than that of the NiTi phase, which was consistent with the order of the phase’s formation calculated. The Ni_3_Ti and NiTi_2_ phases in this study were gradually formed in the Ni/Ti interfacial diffusion layer with the thermal diffusion temperature increased, the absence of NiTi phase may be due to insufficient diffusion time or a small amount of formation [20,21].

The atom diffusion direction in the Cu-Ti binary system could be theoretically analyzed by chemical potential. It is well known that the atom always spontaneously transfers from a high chemical potential to a low chemical potential, and the following theory is summarized by Wu et al. [22]:(4)$$\mu_{i}=\frac{\partial G}{\partial n_{i}}$$

Where $\mu_{i}$ is the chemical potential of component *i*, *G* is the Gibbs free energy, and *n_i_* is the atomic number of component *i*. It can be calculated that the chemical potential of Ti atoms was higher than that of Cu atoms. Under the driving force of the diffusion, the Ti atoms diffused toward the Cu side while the diffusion reaction occurred. Previous researches proved that the Ti atoms with greater diffusion capacity formed Cu_x_Ti_y_ intermetallic compounds on the Cu side [6,23]. Therefore, under the above synergistic effect, the Ti atoms rapidly diffused toward the Cu side, gradually forming the CuTi, Cu_2_Ti, and Cu_4_Ti_3_ intermetallic compounds, which can be expressed as follows [22,24]:(5)$$2\mathrm{Cu}+\mathrm{Ti}={\mathrm{Cu}}_{2}\mathrm{Ti}$$
(6)Cu+Ti=CuTi
(7)$$4\mathrm{Cu}+3\mathrm{Ti}={\mathrm{Cu}}_{4}{\mathrm{Ti}}_{3}$$

Apparently, the intermetallic compounds of Ni_x_Ti_y_ and Cu_x_Ti_y_ were gradually formed in the coating with the annealing temperatures increased. These intermetallic compounds with strong atomic bonds and high hardness served as the reinforcing phases in the diffusion layer, which can effectively improve the tribological properties of the coating [25,26]. 

### 3.2. Tribological Properties of the Ni/Cu/Ni Coating on the Ti-6Al-4V Alloy

It is well known that surface topography is an important factor in describing tribological properties of the coating [27]. As depicted in Figure 5a, the surface of untreated coating was characterized of continuous uniform grain size. As the annealing temperature increased, the coating surface gradually transformed to granulated and densified status (Figure 5b–d). It showed that a large number of fine particles like the Ni_x_Ti_y_ and Cu_x_Ti_y_ phases existed on the surface of the Ni/Cu/Ni coating after the thermal diffusion at 700 °C and 800 °C, which was resulted from a stable diffusion layer with the diffusion reaction between Ni, Cu, and Ti [26].

The microhardness of the annealed coatings was shown in Figure 6a. With the annealing temperatures up to 800 °C, the microhardness progressively increased from 155 to 357 HV. The increased microhardness was ascribed to the formation of the intermetallic compounds and the strengthening effect of solid solution [25,28]. The Cu_x_Ti_y_ and Ni_x_Ti_y_ intermetallic compounds with high hardness can substantially enhance the surface microhardness of the Ni/Cu/Ni coating. At the same time, the annealed coatings had a lower amount of mass loss, and can even be reduced to half of the Ti-6Al-4V substrate. This was mainly because the hard phases dispersed on the coating surface can effectively enhance the coating’s resistance to high contact stresses, thereby optimizing the coating surface wear behavior.

Figure 6b displayed the friction coefficient of the annealed Ni/Cu/Ni coatings. Clearly, the friction coefficient of the Ti-6Al-4V substrate was around 0.6, and it gradually decreased and stabilized at around 0.4 after annealing, which were all composed of the running-in stage and stabilization stage. It was noteworthy that some anomalous peaks appear during the stabilization stage after annealing at 600 °C. This may be related to severe plastic deformation and adhesive wear caused by low microhardness. The process of adhesion, delamination, debonding, and re-stabilization resulted in these anomalous peaks, which implied poor surface tribological properties of the coating. As for the Ni/Cu/Ni coating after annealing at 800 °C, there was no obvious running-in stage and reached quickly to a relatively stabilization stage, which implied the increase of microhardness and the uniform dense diffusion layer can improve the surface tribological properties of the coating. Meanwhile, the plastic deformation ability of the Ni/Cu/Ni coating under the stress concentration was also significantly reduced due to the increase of yield strength and microhardness [29]. Thus, the increase of thermal diffusion temperature was conductive to reducing the friction coefficient of the coating, which was put down to the strengthening effect of the hard particle during the diffusion process.

### 3.3. Wear Mechanism

The Ni/Cu/Ni coating without annealing was susceptible to plastic deformation and adhered to the contact surface to form adhesive wear, which caused continuous cutting penetrated on the coating and wore the Ti-6Al-4V alloy substrate. In this process, a large amount of grinding debris was formed on both sides of the grinding mark, as shown in Figure 7a, which indicated poor wear performance. After annealing, the coating was tightly bonded to the Ti-6Al-4V substrate and formed an effective diffusion layer and distributed a mass of intermetallic compounds, resulting in a prominent improvement in wear performance. The wear marks on the surface of the samples gradually became shallower and flatter (Figure 7b–d), indicating that the adhesive wear gradually weakened. Especially, as the annealing temperatures raised up to 800 °C, only slight wear was caused on the surface of the coating, showing the best tribological properties (Figure 7d).

Obviously, the tribological properties of the coatings on Ti-6Al-4V alloy were improved after annealing, which could be stated using Archard’s law [30,31]:*Q* = *KW/H*(8)
where *Q* is the wear rate, *K* is the friction coefficient, *W* is the applied load, and *H* is the hardness. A low friction coefficient and a high hardness can result in a low wear rate under the same wear conditions. As shown in Figure 8, the wear rate of the Ni/Cu/Ni coating on the Ti-6Al-4V alloy was effectively reduced after annealing. Compared with Ti-6Al-4V substrate, the wear rate was reduced from $1.31\;\times \;10^{-2}$ to $0.28\;\times \;10^{-2}$ mm^3^m^−1^ after annealing 800 °C. It was mainly related to changes in the surface state of the coating. Wang et al. [11] studied that intermetallic compounds can increase the resistance of coatings to high contact stress due to their high microhardness. In addition, the Ni layer and the Cu layer had excellent ductility, improving the brittle fracture ability of the coating. During the adhesive wear process, the Ni layer and the Cu layer with lower hardness first underwent a certain degree of deformation, and the adhesive wear contact area was continuously increased. On the other hand, the Ni_x_Ti_y_ and Cu_x_Ti_y_ intermetallic compounds with high hardness could resist more wear loads and prevented the Ni layer and the Cu layer with low hardness from further deformation. Meanwhile, the good ductility of the Ni layer and the Cu layer can alleviate cracks caused by the loading of the hard-intermetallic compounds. Therefore, the intermetallic compounds with the high hardness and the Ni layer and the Cu layer with good ductility can cooperate with each other to jointly improve the wear resistance of the Ni/Cu/Ni coating.

When the annealing temperature was up to 800 °C, the NiTi_2_ and Cu_4_Ti_3_ phases could be formed in the coating, which had a higher microhardness than the Ni_3_Ti, CuTi, and Cu_2_Ti phases [19,32]. Therefore, they would act as a supporting load to reduce the furrow effect of the counter ball to the coating during the wear test. The diffusion layer with a certain thickness reduced the area of adhesion in the friction surface, weakening the adhesion effect. Especially, the hard Ni-Ti diffusion layer of the Ni/Cu/Ni coating increased the yield strength of the contact surface and effectively reduces the tangential stresses and the interfacial stresses [33,34]. As a result, the adhesion effect and the furrow effect on the surface of the coating can be effectively suppressed by each other after annealing at 800 °C.

## 4. Conclusions

The Ni/Cu/Ni coating were prepared on the surface of Ti-6Al-4V alloy by the electrodeposition and subsequently annealing at 600 °C, 700 °C, and 800 °C for 3 h, respectively. The microstructure and tribological properties of the diffusion layer were systematically investigated. Based on the achieve results the following conclusions can be made:

The Cu, Ni, and Ti atoms in coating displayed complex diffusion behaviors and formed the diffusion layers with a certain thickness. As the annealing temperature increased, the Kirkendall diffusion channel appeared, accelerating the diffusion behaviors of the atoms. In addition, the diffusion layers were mainly composed of the Ni_3_Ti, NiTi_2_, CuTi, Cu_2_Ti, and Cu_4_Ti_3_ intermetallic compounds and the α(Cu, Ni) solid solution with high hardness and strength, which acted as the reinforcing phases of wear resistant materials. The Ni_x_Ti_y_ and Cu_x_Ti_y_ intermetallic compounds were continuously and uniformly distributed in the Ni/Cu/Ni coating, which can tremendously strengthen the surface hardness, reduce the friction coefficient and lessen the wear rate. In wear process, the intermetallic compounds with the high hardness and the Cu layer and the Ni layer with good ductility can cooperate with each other to jointly improve the wear resistance of the coating. The synergistic effect was particularly remarkable at the annealing temperature of 800 °C, showing excellent wear resistance.

## Figures and Tables

**Figure 1 materials-13-00847-f001:**
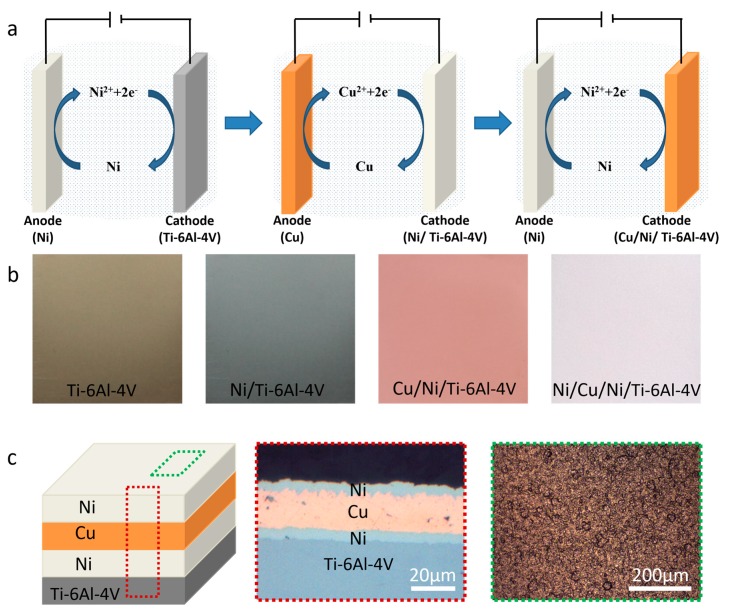
The Ti-6Al-4V electroplated Ni/Cu/Ni coating: (**a**) schematic diagram of preparation technology, (**b**) macroscopic photographs of samples during preparation, and (**c**) schematic diagram and optical microscopy images and cross-sectional microstructure.

**Figure 2 materials-13-00847-f002:**
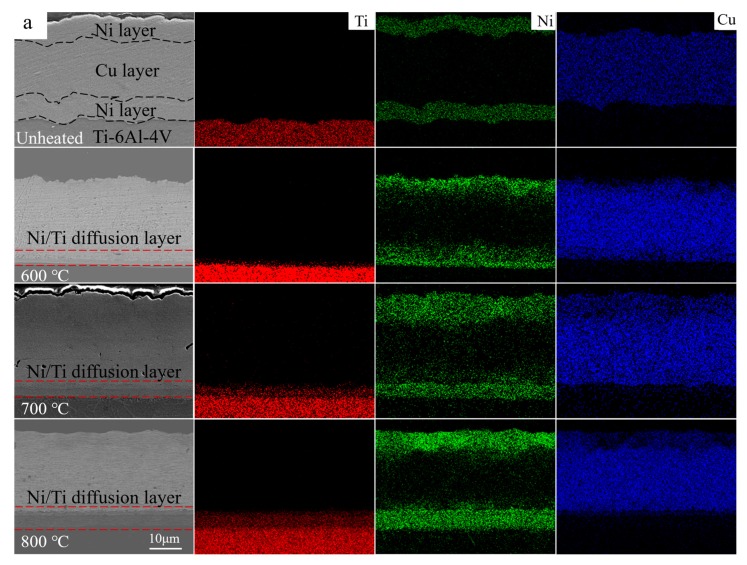
Microstructure of the Ni/Cu/Ni coatings after annealing at various temperatures: (**a**) cross section and corresponding element composition, (**b**) Ni/Ti diffusion layer width, and (**c**) the diffusion layer structure SEM of the Ni/Ti interface for 800 °C. The obvious Ni/Ti diffusion layers were observed in the red line. (**d**) showed the Kirkendall diffusion channel, which was shown in the white area.

**Figure 3 materials-13-00847-f003:**
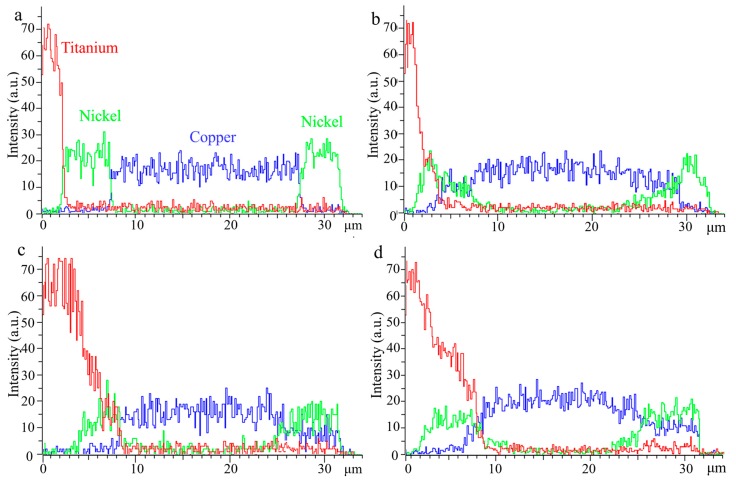
The elements line scan diagrams across the coatings from its substrate to the top after annealing: (**a**) untreated, (**b**) 600 °C, (**c**) 700 °C, and (**d**) 800 °C, respectively.

**Figure 4 materials-13-00847-f004:**
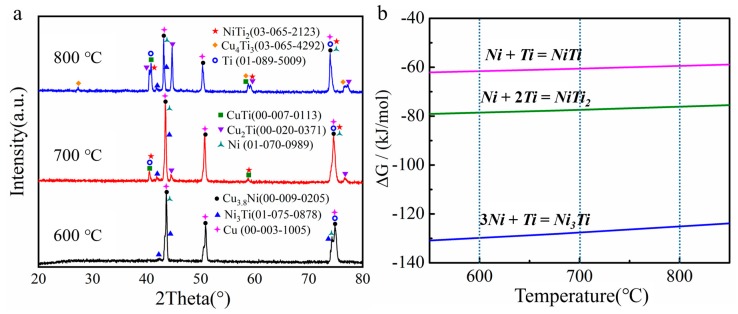
Phases transitions of the Ni/Cu/Ni coatings after annealing at various temperatures: (**a**) surface XRD images and (**b**) Gibbs free energy of the Ni-Ti reaction.

**Figure 5 materials-13-00847-f005:**
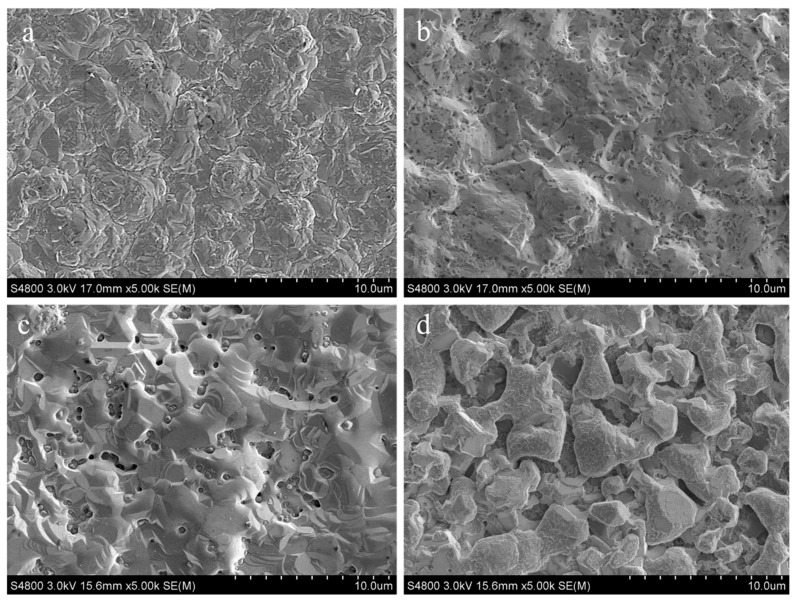
Surface morphology of the Ni/Cu/Ni coatings after annealing at various temperatures (**a**) untreated, (**b**) 600 °C, (**c**) 700 °C, (**d**) 800 °C, respectively.

**Figure 6 materials-13-00847-f006:**
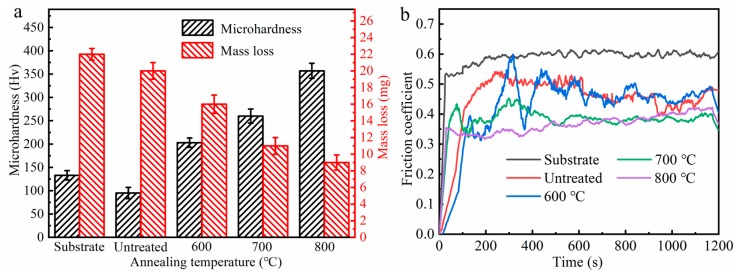
The surface tribological properties of the Ni/Cu/Ni coatings after annealing at various temperatures: (**a**) microhardness and mass loss and (**b**) the change curves of the friction coefficient with sliding time.

**Figure 7 materials-13-00847-f007:**
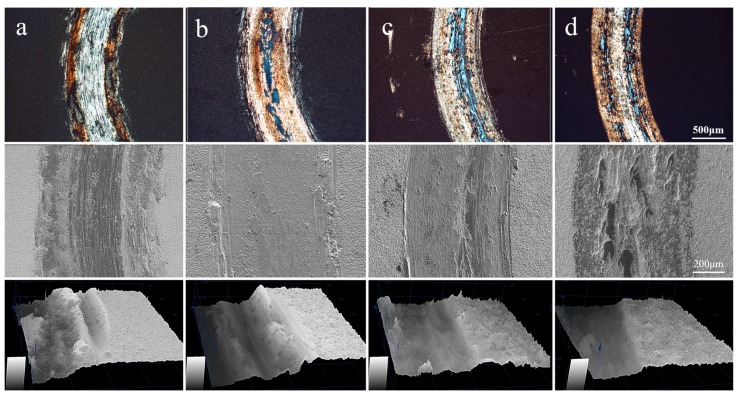
The surface wear morphology of the Ni/Cu/Ni coatings after annealing at various temperatures: (**a**) untreated, (**b**) 600 °C, (**c**) 700 °C, and (**d**) 800 °C, respectively.

**Figure 8 materials-13-00847-f008:**
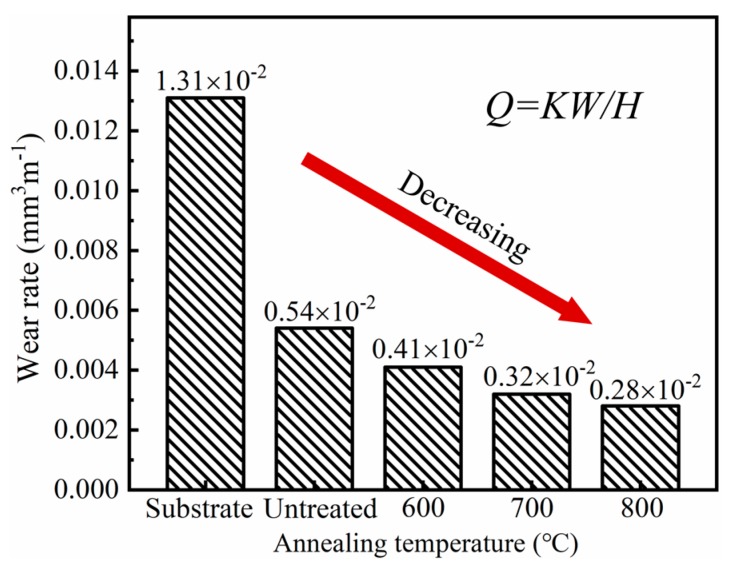
The wear rare of the Ni/Cu/Ni coatings after annealing at various temperatures.

**Table 1 materials-13-00847-t001:** The electrolyte and the operating condition.

	Electrolyte Composition	Operating Condition
Nickel plating	nickel sulfate hexahydrate (NiSO_4_·6H_2_O): 180 g/L sodium sulfate (Na_2_SO_4_): 70 g/L magnesium sulfate (MgSO_4_): 30 g/L sodium chloride (NaCl): 30 g/L boric acid (H_3_BO_3_): 30 g/L	voltage: 3 V temperature: 25 °C duration: 10 min
copper plating	copper sulfate pentahydrate (CuSO_4_·5H_2_O): 210 g/L sodium chloride (NaCl): 20 mg/L sulfuric acid(H_2_SO_4_): 70 g/L	voltage: 0.65 V temperature: 25 °C duration: 20 min

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
