# Peer review of "Tribological Properties of Ni/Cu/Ni Coating on the Ti-6Al-4V Alloy after Annealing at Various Temperatures"

_materials, 2020, doi:10.3390/ma13040847_

Round 1

Reviewer 1 Report

Dear authors,

The paper requires serious corrections to be published in Materials.

I have two sets of comments. The first one is mainly editorial and you can make such corrections easily while the second set as I foresee requires some additional experiments and comprehensive work with literature.

Firstly, the paper title should be corrected: please either use “tribological properties” (in plural) or define exactly what tribological property (friction coefficient, wear resistance, etc.) has attracted your attention mostly.

Please make also corresponding corrections in the Abstract, Keywords, and through the text.

Secondly, I strongly recommend to add more widely known (US, European) alloy and steel grades to Chinese ones in the paper.

So let me propose the following title of the paper: “Tribological properties of Ni/Cu/Ni coating on the Ti-6Al-4V alloy after annealing at various temperatures”.

Experimental section. Please add following details on XRD measurements: anode material, monochromator (no, front, rear), slits.

Please replace “Heat treatment” to “Annealing” in the text and on figures for better clarity.  

I kindly ask you also to check English carefully through the paper.

The second set of comments is connected to your results and their discussion.

I have very serious doubts about elements interdiffusion and resulting phases formation.

1) I am surprised that no Ni, Cu, and Ti lines are observed on XRD patterns (Figure 3(a)). Please explain it. You shall study equilibrium diagrams Ni-Cu, Ni-Ti, Cu-Ti once more to estimate what phases could be found in your system. So I do not expect presence of CuxNi phase because the diagram shows only BCC solid solutions.

The diffraction pattern from the as-deposited system should be also provided.

2)  XRD results are not supported by X-ray microprobe elements distribution data. I suppose that you shall add elements distribution diagrams (also known as linescan diagrams) across the coatings from its top to the substrate after annealings. Probably linescans could reveal better composition of phases formed at annealing.

3) It would be very good to provide high resolution diffraction patterns and linescan elements distribution diagrams as supplementary materials to the paper.

4) It is obvious that annealing affects not only a coating but a substrate too. Please provide hardness of the Ti substrate in the initial state and after annealings because properties of the substrate affect on the tribological behavior of the system.

5) It is necessary to perform similar wear test of uncoated Ti alloy in its initial state and after corresponding annealings to compare friction coefficients and wear resistance of coated and uncoated samples.

Reviewer 2 Report

The aim of this paper is to assess the effect of a thermal treatment on the diffusion and tribological property of the Ni/Cu/Ni electroplated coating, deposited on a TC4 titanium alloy substrate. The topic is compatible with the journal scope. Some modification are required in order to improve its quality and readability.

Concerning the new phases obtained after the thermal teatment a deeper discussion could be planned. At first figure 3a need to be largher enough to allow to better discriminate the obtained phases. Furthermore XRD linear scan of Ti, Ni and Cu element along the cross section could better applied to evidence their distribution. On this concern XRD on grinded surfaces to evidence how change the XRD spectrum at varying distance from the substrate can be applied.

Evaluating figure 4, apparently the roughness observed in d is higher than b. For sack of clarity please add also skewness and kurtosis of the profile. These paramter can be used to better discriminate the differences among the sample's surfaces. These parameters should support the Authors to assess the surface discrepancies in peaks and valley distribution.

Row 199. The stabilization is quite near to 0.4.

To evaluate the wear behaviour of the substrates, heigth profile or mass loss need to be added. The optical 3D scan reported in figure 6 are not suitable to discrimate these info.

The Authors indiates the Archard's law (equation 8) as good index to evaluate the tribological properties of the substrates. The formula 8 was reported, but no results are discussed. Please add a table with wear rate (Q) values for all batches, coupled (of course) with a discussion of the data, or remove equation 8.

In the paper is not well indicated the application of this activity. This aspect need to be well stressed in the introduction. Furthermore well indicate the improvement of Ni/Cu/Ni approach compared to conventional Cu/Ni substrate (already published by the Authors) on the diffusion phenomena and coating performances.

I'm not native english, although a revision to improve its redability is required

Round 2

Reviewer 2 Report

I really appreciate the review work done by the Authors improving all issues highlithed by the reviewer. They have significantly modified the article, deeply improving, in my opinion, its quality.

Consequently, I consider the article suitable for publication in the present form.